# Application of Machine Learning to Metabolomic Profile Characterization in Glioblastoma Patients Undergoing Concurrent Chemoradiation

**DOI:** 10.3390/metabo13020299

**Published:** 2023-02-17

**Authors:** Orwa Aboud, Yin Allison Liu, Oliver Fiehn, Christopher Brydges, Ruben Fragoso, Han Sung Lee, Jonathan Riess, Rawad Hodeify, Orin Bloch

**Affiliations:** 1Department of Neurology, University of California, Davis, Sacramento, CA 95817, USA; 2Department of Neurological Surgery, University of California, Davis, Sacramento, CA 95817, USA; 3Comprehensive Cancer Center, University of California Davis, Sacramento, CA 95817, USA; 4Department of Ophthalmology, University of California, Davis, Sacramento, CA 95817, USA; 5West Coast Metabolomics Center, University of California Davis, Davis, CA 95817, USA; 6Department of Radiation Oncology, University of California, Davis, Sacramento, CA 95817, USA; 7Department of Pathology, University of California, Davis, Sacramento, CA 95817, USA; 8Department of Internal Medicine, Division of Hematology and Oncology, University of California, Davis, Sacramento, CA 95817, USA; 9Department of Biotechnology, School of Arts and Sciences, American University of Ras Al Khaimah, Ras Al Khaimah 72603, United Arab Emirates

**Keywords:** glioblastoma, metabolomic profiling, machine learning, treatment response, concurrent chemoradiation

## Abstract

We here characterize changes in metabolite patterns in glioblastoma patients undergoing surgery and concurrent chemoradiation using machine learning (ML) algorithms to characterize metabolic changes during different stages of the treatment protocol. We examined 105 plasma specimens (before surgery, 2 days after surgical resection, before starting concurrent chemoradiation, and immediately after chemoradiation) from 36 patients with isocitrate dehydrogenase (IDH) wildtype glioblastoma. Untargeted GC-TOF mass spectrometry-based metabolomics was used given its superiority in identifying and quantitating small metabolites; this yielded 157 structurally identified metabolites. Using Multinomial Logistic Regression (MLR) and GradientBoostingClassifier (GB Classifier), ML models classified specimens based on metabolic changes. The classification performance of these models was evaluated using performance metrics and area under the curve (AUC) scores. Comparing post-radiation to pre-radiation showed increased levels of 15 metabolites: glycine, serine, threonine, oxoproline, 6-deoxyglucose, gluconic acid, glycerol-alpha-phosphate, ethanolamine, propyleneglycol, triethanolamine, xylitol, succinic acid, arachidonic acid, linoleic acid, and fumaric acid. After chemoradiation, a significant decrease was detected in 3-aminopiperidine 2,6-dione. An MLR classification of the treatment phases was performed with 78% accuracy and 75% precision (AUC = 0.89). The alternative GB Classifier algorithm achieved 75% accuracy and 77% precision (AUC = 0.91). Finally, we investigated specific patterns for metabolite changes in highly correlated metabolites. We identified metabolites with characteristic changing patterns between pre-surgery and post-surgery and post-radiation samples. To the best of our knowledge, this is the first study to describe blood metabolic signatures using ML algorithms during different treatment phases in patients with glioblastoma. A larger study is needed to validate the results and the potential application of this algorithm for the characterization of treatment responses.

## 1. Introduction


**Key Points:**
➢Specific metabolomic changes are associated with concurrent chemoradiation;➢Metabolomics has the potential to characterize treatment phases in glioblastoma patients;➢In the future, metabolomics may enable the detection of early or distant recurrence.


Glioblastoma is the most common aggressive primary brain tumor in adults and is the leading cause of central nervous system cancer-related death [1]. First-line treatments typically include surgery, radiation therapy (RT), and temozolomide (TMZ) chemotherapy, leading to improvement in 2-year survival rates compared with previous treatments [2]. Despite these treatments, glioblastoma universally recurs, and the overall prognosis remains poor with a 5-year relative survival rate of 6.6% [3]. One of the reasons for tumor relapse is that surviving or dormant tumor cells may acquire new genetic mutations that result in treatment resistance [4]. Many cancer mutations remodel tumor cell metabolism in order to thrive in hostile microenvironments, such as hypoxia, which is typical in glioblastoma [5]. These mutations will cause plasma metabolic changes that are detectable using metabolomic techniques at the time of brain tumor diagnosis [6]. However, little is known about the changes in metabolic profiles in patients after concurrent radiation and TMZ.

Patterns of changes in multivariate data are best uncovered using machine learning (ML) tools. Resulting ML models could then be used to improve the accuracy of tumor diagnosis and choose optimal treatment regimens in the settings of clinical trials that compare different treatment options. Proven to be useful in non-invasive diagnostic approaches for several diseases, such as lung cancers [7,8,9], blood metabolites can be useful markers in assessing a tumor. Therefore, combining ML techniques with metabolomics is a promising strategy for monitoring changes in tumors over time, both in tumor diagnosis and responses to treatment.

In this report, we prospectively enrolled a cohort of patients with isocitrate dehydrogenase (IDH) wildtype glioblastoma, as defined by the new World Health Organization classification, and performed untargeted metabolomics before and after surgery, as well as before and after concurrent chemoradiation. Here, we propose a data-driven approach combining untargeted metabolomics with two classification algorithms to outline specific changes in each stage of treatment in glioblastoma patients. We then screened for changes in the levels of metabolites that were highly correlated during each treatment stage.

## 2. Methods

All 36 patients had histopathologically confirmed diagnoses of IDH wildtype glioblastoma, World Health Organization Grade 4 (WHO Grade 4). The study protocol was approved by the Institutional Review Board of the University of California Davis and written informed consent was obtained. Demographic and clinical data for the study subjects were obtained from a medical record review. Patients received standard-of-care initial treatment with surgical resection, concurrent RT, and chemotherapy. Blood samples were collected before surgery, two days after surgery, prior to starting radiation therapy, and after completing radiation therapy.

### 2.1. Metabolomic Profiling

Untargeted plasma metabolomics via gas chromatography–time of flight mass spectrometry (GC-TOF MS) was performed at the UC Davis West Coast Metabolomics Center. Detailed methods, including plasma extraction and plasma metabolomics, have been reported previously [10,11,12]. In summary, primary metabolites were analyzed using gas chromatography–time of flight mass spectrometry (GC-TOF MS). Samples were extracted with 1 mL of −20 °C cold, degassed acetonitrile:isopropanol:water (3:3:2, *v*/*v*/*v*). In total, 500 μL of supernatant was evaporated to dryness using a CentriVap (Labconco, Kansas, MO). Metabolites were derivatized in two steps as published previously [12]: first, carbonyl groups were protected by methoximation; second, acidic protons were exchanged against trimethylsilyl groups to increase volatility. A 0.5 μL sample was injected with 25 s of splitless time on an Agilent 6890 GC (Agilent Technologies, Santa Clara, CA, USA) using a Restek Rtx-5Sil MS column (30 m × 0.25 mm, 0.25 μm) and 1 mL/min Helium gas flow. Oven temperature was held at 50 °C for 1 min, ramped up to 330 °C at 20 °C/min, and held for 5 min. Data were acquired at −70 eV electron ionization at 17 spectra/s from 85 to 500 Da at 1850 V detector voltage on a Leco Pegasus IV time-of-flight mass spectrometer (Leco Corporation, St. Joseph, MI). The transfer line temperature was held at 280 °C with an ion source temperature set at 250 °C. Standard metabolite mixtures and blank samples were injected at the beginning of the run and for every ten samples throughout the run for quality control. Raw data were preprocessed using ChromaTOF version 4.50 for baseline subtraction, deconvolution, and peak detection. Binbase was used for metabolite annotation and reporting 13.

### 2.2. Statistical Studies

Any missing values were replaced with half-minimum imputation. Data for each compound were then log-transformed for normality and then auto-scaled to make the dataset more Gaussian-distributed and, hence, better suited for univariate analyses. All analyses used linear mixed models with compound intensity as the dependent variable, plasma included as a fixed effect, and the subject included as a random effect. The pre-surgery (0), two days post-surgery (S), pre-radiation (pre-RT), and immediate post-radiation (post-RT) treatment samples were included, and the fixed effect of treatment (0 vs. S) (PR vs. post-RT) was a predictor variable in the model and the effect of interest. The values of interest were the regression coefficients and the associated *p*-value for each compound. The *p*-values were not corrected for multiple comparisons.

### 2.3. ML Modeling: Data Preprocessing and Machine Learning Models

In the present study, we implemented two basic ML algorithms—Multinomial Logistic Regression (MLR) [13] and GradientBoosting (GB Classifier). MLR applies logistic regression to multiclass problems. The GB Classifier combines many weak learning models together to create a strong predictive model with enhanced effectiveness in classifying complex datasets. Data preprocessing was performed on the raw dataset using the StandardScaler module to adjust features to the same scale. The current treatment category values were mapped according to the following scheme: “0—pre-surgery”: 0; “S—post-surgery”: 1; “PR—pre-radiation”: 2; “post-RT—post-radiation”: 3. The parameters for the MLR classifier were C = 100, multi-class = multinomial, maximum iterations = 4000, penalty = 12, and solver = sag. The GB Classifier algorithm was used with the following parameters: learning rate = 0.05, max depth = 8, max features = 16, and number of estimators = 60. To analyze the performance of the proposed algorithms, accuracy, precision, and receiver operating characteristic (ROC) were evaluated to validate the predictive ability of the models.

The accuracy, precision, recall, and ROC AUC scores were calculated for the test set using the scikit-learn library in python [14]. The accuracy score was computed using the sklearn.metrics.accuracy_score () function, which evaluates the true labels according to the following equation, which is defined as
$$\mathrm{accuracy}\left(y,{\hat{y}}_{i}\right)=\frac{1}{n_{\mathrm{samples}}}\overset{\left\{\mathrm{nsamples}-1\right\}}{\underset{\left\{i=0\right\}}{\sum }}({\hat{y}}_{i}=y_{i})$$
where ${\hat{y}}_{i}$ is the predicted value of the i-th sample, and y_i_ the corresponding true value.

The precision was computed using the sklearn.metrics.precision_score function and defined as the ratio of true positives/(true positives + false positives). Recall was calculated using the sklearn.metrics.recall_score function and defined as the ratio of true positives/(true positives + false negatives).

Finally, area under the receiver operating characteristic curve (ROC AUC) scores were computed using the sklearn.metrics.roc_auc_score function, which were then compared between the LR and GB models and also compared with the ideal value of 1 [15]. The confusion matrix was used to visually describe the accuracy of the different models in identifying the treatment status, including the actual vs. predicted values. Machine learning models were created in Python (version 3.6) installed through Anaconda and using the scikit-learn package [16] on a computer equipped with an Intel (R) Core™ i7-10700T CPU @ 2.00GHz.

## 3. Results

### 3.1. Patients

In total, 36 patients with glioblastoma, IDH wildtype, WHO Grade 4 (determined via immunohistochemistry), were identified at the UC Davis Neuro-Oncology clinic (18 MGMT methylated, 16 MGMT unmethylated, 2 MGMT status unknown). There were 21 males and 15 females, the median age at diagnosis was 63.5 years, the median BMI at diagnosis was 28, and 30 patients self-identified as white. All patients underwent surgical intervention at our institution. The number of available samples per stage of treatment was: 0: 36, S: 32, PR: 28, and post-RT: 17 (Table 1). Two patients did not receive RT or TMZ due to poor clinical conditions and functional status.

### 3.2. Metabolomic Changes

We identified 157 unique metabolites with a retention index and mass spectral matching [12]. Compared with pre-surgical samples, the two days post-surgical plasma samples showed significant increases in aromatic amino acids using *p*-values, including tryptophan, phenylalanine, and tyrosine, among other metabolites (Figure 1), while sugar alcohols and lipid metabolites showed significant decreases post-surgery, including sorbitol, mannitol, glucuronic acid, mannose-6-phosphate, glycerol, 6-deoxyglucitol, and gluconic acid, as well as oleic acid, linoleic acid, linolenic acid, 3-hydroxybutyric acid, and gamma-tocopherol (Figure 1 and Appendix A).

When comparing pre-radiation vs. post-radiation samples (Figure 2), significant increases were noted in specific amino acids, including glycine, serine, threonine, and oxoproline, in addition to sugar alcohols such as xylitol, 6-deoxyglucose, gluconic acid, glycerol-alpha-phosphate, ethanolamine, arachidonic acid, linoleic acid, propyleneglycol, and triethanolamine, along with unsaturated fatty acid, TCA metabolites, succinic acid, and fumaric acid. A significant decrease post-radiation was only noted for 3-aminopiperidine-2,6-dione (pyroglutamine) (Table 2).

## 4. ML Models for Classifying Treatment Stage

In total, 105 samples were randomly divided with 70% of the data allocated to the training dataset and 30% allocated to the testing dataset. Between the two established models, the accuracy of MLR was 0.78, followed by the GB Classifier with 0.75. The ROC AUC score values were 0.89 for MLR and 0.91 for the GB Classifier. The highest precision score was for the GB Classifier (0.77), followed by MLR (0.75). The highest recall score was for MLR (0.78). (Figure 3).

The 2 models correctly classified 12 out of 12 patient samples during the pre-surgery stage, as shown in the confusion matrices of these models on 32 samples of the test data. Since our goal was to develop algorithms that are able to classify a set of metabolomics data into one of four treatment stages, we opted to use a 4×4 confusion matrix to determine which of the model classifications were right or wrong. Pairwise comparisons reflected a diagonally populated confusion matrix, indicating that our models were capable of identifying correct treatment stages. The performance scores (Figure 3E) were close between the two models. The MLR algorithm correctly classified four out of five samples for the post-surgery stage. The GB Classifier, on the other hand, classified five out of five samples for this stage. For pre-radiation and post-concurrent chemoradiation therapy samples, the MLR Classifier algorithm showed the most accurate classification of pre-radiation vs. post-radiation, where three out of six samples were correctly classified by MLR model versus two out of six by the GB model (Figure 3).

## 5. Correlation Analysis

Identifying characteristic metabolite patterns associated with the clinical stages can provide helpful insights that can be used together with prediction models to come up with better decisions. In addition, this approach could identify changes in hub metabolites independent of the model prediction. We decided to focus on metabolites that are strongly correlated for the following reasons: first, strongly correlated metabolites are likely to be in the same metabolic pathway, and, secondly, we wanted to simplify the metabolic profile per clinical stage by selecting a subset of detected metabolites.

To forecast a pattern of metabolite changes per clinical stage, specifically during tumor presence (pre-surgery), post-surgery, pre-radiation, and post-radiation, we computed pairwise correlations using Pearson correlation with a particular cut-off (positive pairs r > 0.90). The result was twenty-one metabolites shown in a heatmap (Appendix A). Comparing pre-surgery to post-radiation showed a similar pattern of changes to that in pre-surgery and post-surgery (Appendix A). Comparing pre-radiation to post-radiation for the 21 metabolites showed a significant decrease in linoleic acid.

Next, we investigated specific patterns for metabolite changes in the 21 selected metabolites. Comparing pre-surgery to post-surgery samples, the results showed a significant increase, *p* < 0.0001, for deoxy-pentitol, deoxytetronic acid, fucose, deoxyglucose, isoleucine, leucine, lactose, and lactulose; *p* < 0.01 for fumaric acid, heptanoic acid, succinate semialdehyde, indoxyl sulfate, and mandelic acid; and *p* < 0.05 for caproic acid and threonic acid and a significant decrease, *p* < 0.0001, for linolenic acid, linoleic acid; *p* < 0.01 for gluconic acid; and (*p* < 0.05) for gluconic acid lactone.

Based on the heatmap correlations (Appendix A), few metabolites from the filtered 21 metabolites showed high correlations. Among these were the correlations between deoxypentitol and lactulose, lactose, ribitol, arabitol, 2-dexosytetronic acid, and fumaric acid (Figure 4A,B). This could suggest a correlation with the pentose phosphate pathway, especially with increased levels of nucleic acids noted after surgery (Figure 1).

Comparing post-surgery to pre-radiation, there was a significant increase (*p* < 0.05) in deoxypentitol, heptanoic acid, leucine, and isoleucine, while there were no significant changes for other metabolites (Appendix A). Comparing post-surgery to post-radiation, a significant increase, *p* < 0.01, was found for deoxy-pentitol and *p* < 0.001 for arabitol (Appendix A).

## 6. Discussion

In this study, we demonstrated specific plasma metabolomic changes associated with surgery and chemoradiation treatment in patients with pathologically confirmed IDH wildtype glioblastoma. Our study compared two ML algorithms using samples from patients diagnosed with glioblastoma and found that an MLR algorithm can better classify the phase of treatment in post-operative patients in training and testing groups.

The first-line treatment for patients with glioblastoma is typically surgery, RT, and chemotherapy. Patients with suspected brain tumors discovered on imaging are usually referred for surgical intervention because this provides tumor tissue for histological diagnosis and molecular testing, and the extent of resection is an important prognostic factor. While glioblastoma tumors vary between patients in their molecular features, similar treatments have been applied to the majority of these tumors due to the lack of effective and specific treatment options. Although MRI is the best non-invasive technique to evaluate responses to treatment, it usually carries limited value shortly after radiation therapy. There is no set of imaging features that can fully predict tumor responses to treatment with a high level of accuracy, and the interpretation of these features is not easy to standardize. This is particularly relevant in the context of post-radiation scans. Little is known about the metabolomic changes associated with early treatment stages such as surgery and chemoradiation. In these situations, it would be useful if metabolomics in the future could help assess tumor response to treatment.

Our study compared prospectively collected blood metabolomic profiles obtained from glioblastoma IDH wildtype patients before and after surgical resection as well as before and after concurrent RT with TMZ chemotherapy. A metabolite-by-metabolite analysis of the pre-radiation vs. post-radiation samples led to the detection of 15 metabolites that increased after radiation therapy, while 1 metabolite decreased. Obtaining a metabolomic profile for patients may help in the improvement of treatment strategies, and several compound classes such as sugar alcohols were previously detected as markers in other cancers [10]. Glycine and serine charge methyl-donors in one-carbon metabolism [17], a critical pathway in nucleotide metabolism via SAM, folate, and methionine cycles [18]. Ethanolamine and glycerol-alpha-phosphate are building blocks for lipid metabolism, and they reportedly shift the equilibrium from ethanolamine to phosphoethanolamine reported in breast cancer [19]. An increase in 6-deoxyglucose and gluconic acid is an interesting observation, especially when coupled with observed decreased gluconic acid post-surgery. Along with other sugar alcohols, these metabolites may indicate nonclassical glucose utilization, as also observed in liver cancer and kidney cancers [10,20,21].

Cancer treatment (surgery, radiation therapy, and chemotherapy) in our study samples altered the levels of both endogenous and exogenous compounds. The endogenous changes can be rationalized as typical cancer-related compounds, including glycine/serine (for changes in methylation status via glycine dehydropenase and one-carbon metabolism), lipid metabolism that responds to cell replication (via the backbones of complex lipids, such as glycerol-phosphate and ethanolamine, but also arachidonic acid and linoleic acid), and TCA cycle alterations (via succinic and fumaric acid, highlighting the importance of succinyl COA-efflux for biosynthetic purposes). In addition, 3-aminopiperidine 2,6-dione is an isomer of methylhydrouracil and dihydrothymine, pointing to alterations in nucleoside biosynthetic pathways. Deoxyglucose, however, might be interpreted as an overflow of oxidative pathways aberrant to glycolysis; a more likely explanation is that these metabolites are hexose. Further confirmation of some of these metabolites is required, while a few exosome compounds, specifically, propyleneglycol, triethanolamine, and xylitol, may stem from altered gut microbials due to chemoradiation stress. Having said that, the full implication of exogenous metabolites in treatment or treatment stage is not fully clear.

In two recent studies [4,22], approximately 20% of mutations and copy-number variants generated were new in tumor samples obtained from recurrent tumors (surgery #2) versus the initial diagnosis (initial surgical resection). Little is known about the underlying metabolic alterations that accompanied these mutations or how they may promote either tumor recurrence or resistance to treatment in patients with glioblastoma. Metabolomics can be used to broadly detail metabolic reprogramming in the fields of neuro-oncology, clinical diagnostics, and prognostics [23]. Chinnaiyan et al. reported a metabolomic profile in tumor tissue (blood was not assessed): phosphoenolpyruvate accumulation and decreased pyruvate kinase were highly correlated with an aggressive subgroup of high-grade glioma [24]. Of note, pyruvate kinase activity was measured here using a colorimetric assay. In addition, a retrospective study identified a unique plasma metabolomic profile that was predictive of prognosis in glioblastoma patients [25], reporting that higher levels of methionine and arginine were associated with a 37% and 34%, respectively, increased probability of two-year survival, while increased levels of kynurenate were associated with a 55% decreased probability of two-year survival. Furthermore, in another study, a metabolomic profile was reported to correlate with glioblastoma versus lower-grade glial tumor phenotypes [26]. A recent review of 47 metabolomic profiling studies pointed out that the absence of prospective longitudinal metabolomic studies, such as our study, has impeded the realization of the full potential of metabolomic profiling in cancer diagnosis and prognosis [27].

Based on the correlation analysis, we sought to study selected metabolites positively correlated by more than 90%. Our decision for this rigorous correlation cut-off was based on our interest in identifying a pattern of change in specific metabolites across three clinical stages: pre-surgery, post-surgery, and post-radiation. We focused on metabolites that significantly change in pre-surgery vs. post-surgery to study the pattern of change in these metabolites post-surgery (tumor-free) and compare this pattern to later stages during treatment, specifically, post-radiation. This could explain, at least in part, the limited overlap between our findings and previous studies, as those studies concentrated on the later stages of the disease. We reported the highest correlations between deoxypentitol and lactulose, lactose, ribitol, arabitol, 2-dexosytetronic acid, and fumaric acid (Figure 4A,B). This, along with increased levels of nucleic acids noted after surgery, could suggest the correlating involvement of the pentose phosphate pathway, which is known to be involved in glioblastoma [28]. We argue that the metabolites that significantly change post-surgery reflect a tumor-free condition, and a similar pattern in comparing post-radiation to pre-surgery will suggest a positive prognosis, while deviation from this pattern could suggest relapse or a negative prognosis.

The generalization of our prediction models is limited by the small sample size and the collection of the samples being restricted to one center. An additional external dataset from a broader geographical area would limit model bias and allow us to evaluate model generalization. Furthermore, the study is missing control groups consisting of samples from normal donors. However, comparing metabolites pre-surgery, post-surgery, pre-concurrent chemoradiation, and post-concurrent chemoradiation longitudinally in the same patient is considered to be using each patient as his/her own control in a prospective fashion.

Our prediction models do not include features of BMI, race, age, or gender. Although these could play a significant role in recurrence, our study strongly argues for the possibility of developing more general models that can predict responses to treatment in various groups of people. It is also important to note that, although our developed algorithms showed prominent performance in classifying patients in each treatment group, it remains unclear if these algorithms will show similar performance in predicting treatment responses or tumor progression. It remains possible that selecting specific metabolites based on correlation analysis will rule out other metabolites that are important to the classification process.

## 7. Conclusions

In our patient population with glioblastoma *IDH wildtype*, we identified several plasma metabolite changes that could be associated with surgery, radiation, and chemotherapy. The validation of these results in a larger cohort and a further investigation of the underlying mechanism of these metabolites is still needed. Future validation may provide a guide for monitoring responses to treatment. In this study, we designed and developed a Multinomial Logistic Regression algorithm that could classify treatment phases in glioblastoma patients undergoing surgery and concurrent chemoradiation.

## Figures and Tables

**Figure 1 metabolites-13-00299-f001:**
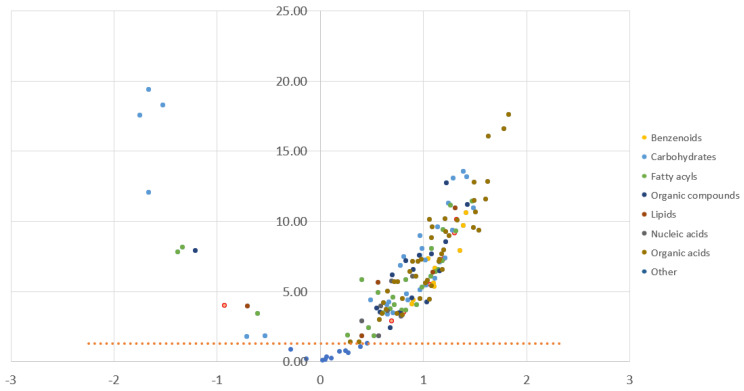
Volcano plot prepared from metabolites, displaying the comparison between pre-surgery samples, “D0”, and post-surgery samples, “S”; significance threshold *p* < 0.05. A positive value (right upper quadrant metabolites above dotted red line) means that “S” is higher than “0” (metabolite increased after surgery). A negative value (left upper quadrant metabolites above dotted red line) means that “S” is lower than “0” (metabolite decreased after surgery). The *X*-axis is a standardized regression estimate (beta); the *Y*-axis is -Log10*P*. Colors displayed on the right reflect group metabolites by superclass. “Organic compounds” include organic nitrogen compounds, organic oxygen compounds, and organoheterocyclic compounds. “Lipids” include prenol lipids, glycerolipids, and sterol lipids. “Other” includes metabolites with a lower match score or library match (3-aminopiperidine-2,6-dione, 9-myristoleate, succinate semialdehyde, and phosphate (increased after surgery)). Detailed metabolites with significant changes can be found in Appendix A.

**Figure 2 metabolites-13-00299-f002:**
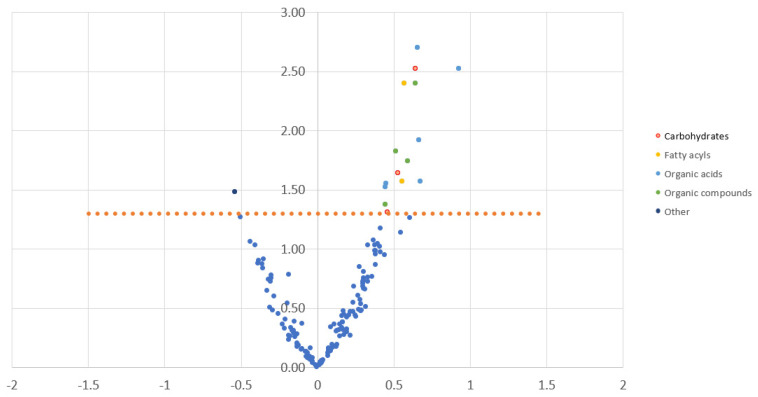
Volcano plot prepared from metabolites, displaying the comparison between pre-radiation samples, “PR”, and post-radiation samples, “post-RT”; significance threshold, *p* < 0.05. A positive value (right upper quadrant metabolites) means that “post-RT” is higher than “PR” (metabolite increased after radiation). A negative value (left upper quadrant metabolites) means that “post-RT” is lower than “PR” (metabolite decreased after radiation). The *X*-axis is a standardized regression estimate (beta); the *Y*-axis is -Log10*P*. Colors displayed on the right reflect group metabolites by superclass. “Organic compounds” include organic nitrogen compounds, organic oxygen compounds, and organoheterocyclic compounds. “Other” includes one metabolite with a lower match score or library match (3-aminopiperidine-2,6-dione (decreased after chemoradiation)). Detailed metabolites with significant changes can be found in Table 2.

**Figure 3 metabolites-13-00299-f003:**
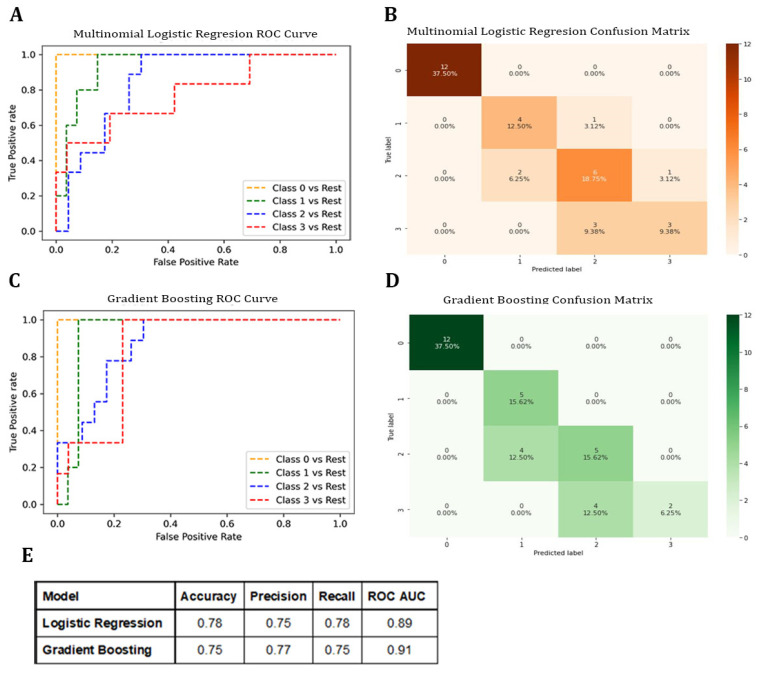
Machine learning algorithm for the classification of clinical stages based on untargeted metabolomics. (**A**,**B**) The Multinomial Logistic Regression model: “pre-surgery”: 0; “post-surgery”: 1; “pre-radiation”: 2; “post-radiation”: 3. (**A**) The receiver operating characteristics (ROC) curve. (**B**) The confusion matrix when tested on the test dataset. (**C**,**D**) The Gradient Boosting model: (**C**) the receiver operating characteristics (ROC) curve; (**D**) the confusion matrix when tested on the test dataset. (**E**) Model performance comparison: Comparison of multinomial regression and Gradient Boosting models based on the accuracy, precision, recall, and ROC AUC score of the testing dataset.

**Figure 4 metabolites-13-00299-f004:**
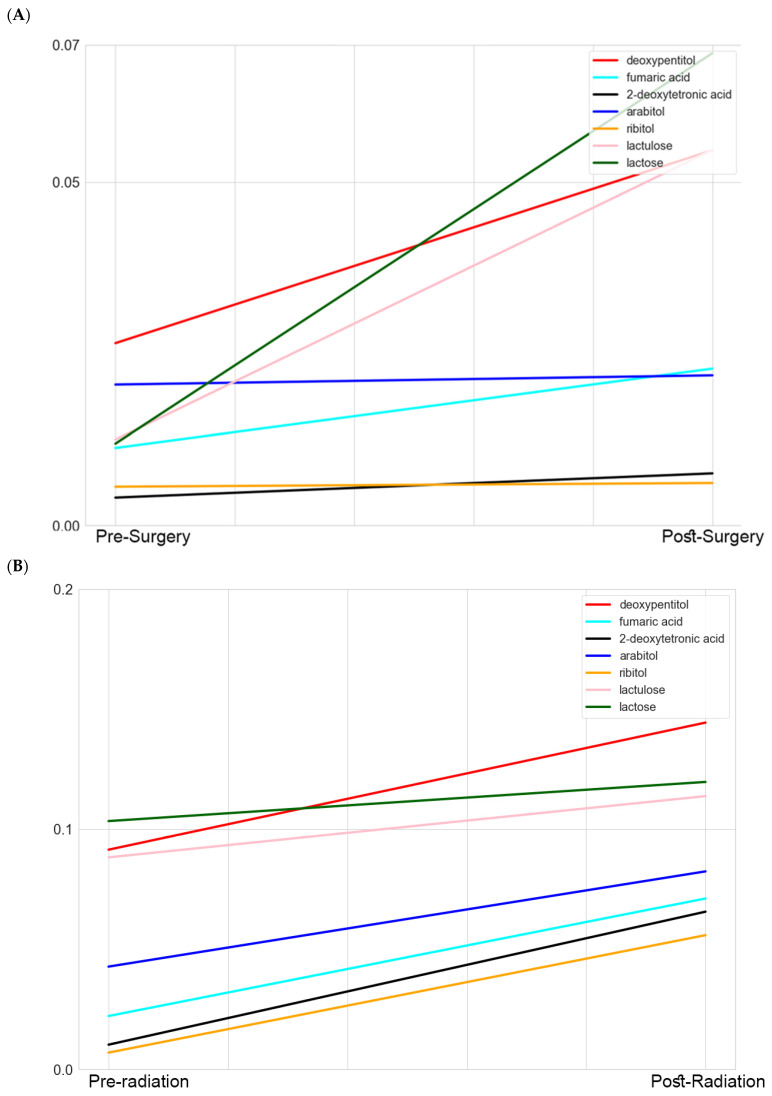
Comparative profiling of highly correlated metabolites per treatment stage. The lines represent the mean of the scaled data for each metabolite. (**A**) Comparison of levels of highly correlated metabolites in pre-surgery vs. post-surgery. (**B**) Comparison of levels of highly correlated metabolites in pre-radiation vs. post-radiation. This figure demonstrates selected metabolites with the highest correlations from a total of 21 metabolites. A full list is provided in Appendix A.

**Table 1 metabolites-13-00299-t001:** Demographics of patients: “0”, pre-surgery sample; “S”, two days post-surgery sample; “PR”, pre-radiation sample; and “RT”, immediate post-radiation sample.

Patient #	Sex	Ethnicity	Age at Dx yr	BMI at Dx	0	S	PR	RT
1	M	White	60	40	X	X	X	X
2	M	White	72	30	X	X	X	
3	M	Hispanic	43	28	X	X		X
4	M	Asian	49	57	X	X		X
5	F	White	78	23	X	X		
6	M	Hispanic	65	22	X	X		X
7	M	White	72	41	X	X	X	
8	M	White	80	24	X	X	X	X
9	F	White	61	27	X	X	X	
10	F	White	69	25	X	X	X	
11	M	Indian	60	27	X		X	X
12	F	White	61	25	X	X	X	
13	F	White	52	27	X	X		
14	M	White	62	30	X	X	X	
15	M	White	69	31	X	X	X	X
16	M	White	67	44	X	X		
17	F	White	82	28	X	X	X	
18	F	White	55	29	X	X		
19	M	African American	47	37	X	X	X	X
20	M	White	63	30	X	X	X	X
21	F	White	86	27	X	X	X	
22	F	White	64	31	X	X	X	X
23	M	White	56	22	X	X	X	X
24	F	White	69	26	X	X	X	X
25	F	NA	69	27	X	X	X	
26	M	White	64	36	X	X	X	X
27	M	White	68	28	X		X	X
28	M	White	69	28	X	X	X	X
29	F	White	58	27	X		X	X
30	F	white	66	27	X	X	X	
31	M	White	55	28	X	X	X	X
32	F	White	60	20	X	X	X	
33	M	White	58	28	X	X	X	
34	M	White	53	30	X	X	X	
35	M	White	58	26	X	X	X	
36	M	White	76	35	X			

**Table 2 metabolites-13-00299-t002:** Detailed metabolites with significant changes. Subclass information obtained using Metabolomics Workbench (https://www.metabolomicsworkbench.org/databases/refmet/index.php) (accessed on 26 December 2022). “NA”, lower match score or library match.

BinBase Name	PubChem	Superclass	Subclass	rq _est	*p*-Value
3-aminopiperidine-2,6-dione	134508	NA	NA	−0.54	0.033
succinic acid	1110	Organic acids	TCA acids	0.45	0.03
fumaric acid	444972	Organic acids	TCA acids	0.67	0.027
threonine	6288	Organic acids	Amino acids	0.45	0.028
glycine	750	Organic acids	Amino acids	0.66	0.002
serine	5951	Organic acids	Amino acids	0.66	0.012
glycerol-alpha-phosphate	754	Organic acids	Organic phosphoric acids	0.93	0.003
xylitol	6912	Carbohydrates	Sugar alcohols	0.53	0.023
6-deoxyglucose	441480	Carbohydrates	Hexoses	0.64	0.003
glucuronic acid	94715	Carbohydrates	Sugar acids	0.46	0.049
linoleic acid	5280450	Fatty acyls	Unsaturated FA	0.57	0.004
arachidonic acid	444899	Fatty acyls	Unsaturated FA	0.56	0.027
ethanolamine	700	Organic nitrogen compounds	1,2-aminoclcohols	0.45	0.042
triethanolamine	7618	Organic nitrogen compounds	Tertiary amines	0.59	0.018
propyleneglycol	259994	Organic oxygen compounds	1,2-diols	0.52	0.015
oxoproline	7405	Organoheterocyclic compounds	Pyrroline carboxylic acids	0.65	0.004

## Data Availability

Data is not publicly available due to privacy.

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
