# Peer review of "Application of Machine Learning to Metabolomic Profile Characterization in Glioblastoma Patients Undergoing Concurrent Chemoradiation"

_metabolites, 2023, doi:10.3390/metabo13020299_

Round 1

Reviewer 1 Report

General Comments:

The authors present the application of two machine learning methods to a metabolomics dataset from a set of glioblastoma patients.  The purported goal as described in the abstract is to “predict treatment phases”.  The notion of prediction versus classification should be clarified.  The ML methods demonstrate the ability to correctly identify which stage a sample belongs.  In the latter part of the abstract, it is stated that the algorithms could predict treatment responses.  It does not appear that this dataset considered response to treatment, rather it characterized the metabolomic features of each stage.  An analysis more in line with the pharmacometabolomics approach whereby pre-surgical metabolic profiles could predict treatment outcomes would be required to claim predictive value.  It appears that all patients received the same treatment course comprised to surgical resection, along with radiation and chemotherapy.

The metabolites that were found using the ML models are somewhat random including many exogenous compounds.  This paper then applies a correlation analysis to look for patterns, but again, no meaningful patterns arise.  Perhaps with a significant improvement in the data visualization approaches the meaning could become clearer.  

Specific Details:

Section 2.2 Statistical Methods:  It would be useful to know how the ML modeling was carried out e.g. with what specific software, programming language, packages, etc.

Section 3.2 Metabolite changes:  The paper describes a number of metabolite changes, but some further review of the identity of these metabolites should have been carried out.  Many of the significantly altered metabolites are most likely exogenous contaminants and dietary compounds including the sugar acids such as sorbitol, mannitol, xylitol, 6-deoxyglucose and 6-deoxyglucitol. 

On page 5 line 167, the metabolites arachidonic acid and linoleic acid are incorrectly included in a list of TCA cycle intermediates.

Figures 1 and 2 would benefit from a clearer description of how this was generated and the meaning of the beta value.  The annotations on the dots are not readable.  Removing the annotations, compressing the figure and including the information in supplementary Tables 1 and 2 next to the plot would be useful.

Figure 3 is not well described and its relevance to the goals of the project are unclear.  Without an improved description, this should be removed.

Section 3.2.1. ML Models for predicting treatment stage:  This section would be improved with added description of how the accuracy, precision and recall scores were derived. This paper demonstrates and interesting application of a 4x4 confusion matrix.  Many readers of Metabolites may only be familiar with the more common 2x2 so some additional description would be useful.

3.2.2 Correlation Analysis:  It is not clear if all of the data was used in the correlation analysis or if there was any application of the ML methods in selecting metabolites.  Beyond showing the heatmap, the authors do not seem to make any attempt to identify clusters of significantly altered metabolites. 

The color gradient on the map makes it very difficult to interpret.  The use of a gradient going from -0.4 to 1 is unusual.  It is more typical to have low correlations near zero to white and positive and negative correlations presented in contrasting colors e.g., red and blue.  This figure should be redrawn for clarity or perhaps simply removed as it does not seem to contribute significant information.

Figure 6.  This plot and the color bar plots on page 10 are very difficult to interpret.  Many of the colors are very similar and so it’s difficult to make a comparison between the stacked bars e.g. gluconic and caproic acid both look black, lactulose and deoxpentiol are both a similar reddish color.  There are other more effective visualization approaches that would make this data more interpretable.  A simple line plots where the metabolites are along the x-axis and each group (O, S PR and PostRT) has a line that goes across.  There may be a lot of overlap, but the significant features would stand out.  A ribbon chart may also be useful.

Section 4: Discussion: The discussion leads off with the notion that the ML algorithms can predict the status of tumor progression.  The algorithms did show the ability to classify which patients were in each treatment group, but this is not the same as predicting tumor stage or progression.

There is some discussion of specific metabolites in the discussion, but some acknowledgement that many of these are exogenous components and unlikely to contribute to a clinical diagnostic should be included.   While it is possible that 6-deoxygluocse could signal a completely novel glucose utilization pathway, it is very unlikely, and it would be prudent to run a quick spiking experiment to confirm this annotation.  

Page 12, first paragraph.  This paragraph describes earlier metabolomics studies, but there is very little overlap with the findings of this study.  Overall, the significant metabolite changes in this study are quite diffuse and do not seem to point to any specific pathways.

Page 12, line 279: “: phosphoenolpyruvate accumulation and decreased pyruvate kinase was highly correlated with an aggressive…”.  This sentence appears to indicate that pyruvate kinase is a metabolite.  The decrease in the enzyme PK should be clarified including how this was measured e.g. PK activity, PK gene expression, Western…

Author Response

Thank you

Reviewer 2 Report

The authors have presented a very interesting area of research regarding Machine Learning as a tool for the Characterization of Metabolomic Profiles in Glioblastoma Patients, undergoing Concurrent 3 Chemoradiation. 
This is really a very interesting area of research and may support a lot of diagnoses and treat patients accordingly. 

Although I have some concerns, owing to my curiosity-
(i) Authors should focus on the research gaps in this area adequately. 
(ii) What are the benefits of ML tools wrt the Regression model in this regard?
(iii) This paper seems to be like a data analysis paper later than a clinical paper. In my opinion, I believe that authors should also focus on different aspects of chemical significance related to these metabolites. 
(iv) I am also interested to know about how these people deal with different mutations and diversity of glioblastoma patients.
(v) from a statistical point of view, I believe that the number of patients is relatively quite low and it also includes a lot of diversity among such as sample-pool for example; some are males, females, white, Hispanic, RT/TMZ treated & non-treated, MGMT unmethylated, methylated and of unknown status. So, I have a doubt that such a small sample pool may adequately address such a great diverse range/of patients with variables. In my opinion, authors should focus on a uniform range of samples with lesser diversity.  The inclusion of positive and negative controls would also be a better approach.

(vi) I couldn't able to find whether the authors validated the final model with some independent samples which were not used in the above testing?

Author Response

Thank you

Reviewer 3 Report

This manuscript reports the application of machine learning methods to analyze metabolite profiles in glioblastoma patients undergoing surgery and concurrent chemoradiation. Many data analysis techniques have been used in this study, including volcano plot, correlation analysis, comparative profiling. Multinominal logistic regression and GB classifier are exploited to predict treatment categories using metabolite profiles. Although it is an interesting research, the data presentation and discussion are poorly structured. This manuscript does not reach the required quality standard of metabolites.

Minor points:

1. Page 2 Line 72. In both introduction and conclusion authors discussed the correlation between cancer mutations with plasma metabolic changes. Is it relevant to this study? For example, how could probe the mutations in these glioblastoma patients based on analysis you have done in the manuscript?

2. Page 3 Line 107. Why did you do log transformation? What do you mean by auto scale?

3. Page 8 Line 198. The (A) (B) captions are repeating. There is no panel E in the figure.

4. Page 8 Line 212. Here it indicates the Pearson correlation analysis, however, Spearman’s correlation is mentioned in the caption (page 9 Line 230). Please clarify.

Major issues:

5. Page 4 Line 149. This is a key experiment which sets the foundation for the whole manuscript, however, no details are provided. Authors should add a section for the GC-MS experiments, provide representative experimental data, and reveal the metabolite identification protocol.

6. Page 5 Line 169. Table 2 is not in the manuscript.

7. Page 7 Line 179. Figure 3 clustering is not described in the manuscript.

8. Page 7 Line 182. A validation set should be split for tuning machine learning hyperparameters.

9. Page 7 Line 190. The post-surgery stage prediction is discussed here. However, both “pre-radiation” and “post radiation” predictions are pretty off, how to interpret it?

10. How do you compare the multinomial logistic regression model with GB model?

Author Response

Thank you

Round 2

Reviewer 1 Report

see attached

Author Response

Please review attachment 

Thank you 

Reviewer 3 Report

In this revised manuscript, the authors attempted to address the questions raised from the reviewers. They have corrected and reorganized the data for the completion of the paper. However, authors depict the experiment and data analysis results without providing in depth discussions. For example, in Figure 4 authors demonstrate the classification performance of 2 models, although the classification accuracy differences for different treatment stages and the model comparison & diagnostics are not explained. In addition, there is a lack of detailed descriptions for techniques, such as ChemRich metabolite grouping. I feel that this work is still not a good fit for a high-quality journal like metabolites.

Author Response

Thank you 

Round 3

Reviewer 1 Report

Thanks for the careful attention to the suggestions.  The paper is significantly improved.  I would like one final adjustment to line 331.  The idea that a deoxyglucose compound could come from "overflow of oxidative pathways aberrant to glycolysis" is interesting, but I can't find any support for this notion.  Nor is it clear what this actually means.  If you truly think that this is a correctly identified metabolite then this should be supported by some reference indicating how it could be made - I have found none.  

Another option would be to explain that this is very likely a hexose, but the library match may not be correct and further confirmation on some of this and some other unusual metabolites is required.  As I mentioned in my last review it would be interesting to see if there were other annotations for this metabolite that has similar matching metrics.  

Author Response

The point raised by the reviewer is well taken, edits to the discussion section were made. 

We truly thank the reviewers and especially reviewer 1 for their commitment to the improvement of our manuscript. 

Thank you! 

Reviewer 3 Report

Authors have addressed some of my previous concerns in the revised manuscript.

Author Response

We thank the reviewers again for their constructive feedback.